# OpenReview forum: "STARE: Step-wise Temporal Alignment and Red-teaming Engine for Multi-modal Toxicity Attack"
_ICML.cc/2026/Conference — ICML 2026 regular_

### Official Review · Reviewer_98L2 · 2026-03-06

**Soundness:** 3
**Presentation:** 3
**Significance:** 2
**Originality:** 2
**Overall Recommendation:** 5
**Confidence:** 3

**Summary:**

This paper introduces STARE, a hierarchical reinforcement learning (RL) framework designed for multi-modal toxicity attacks. The framework combines high-level prompt editing with low-level LoRA fine-tuning of Text-to-Image (T2I) models, treating the denoising trajectory of diffusion models as a structured attack surface. A central contribution of the work is the discovery of the "Optimization-Induced Phase Alignment" (OIPA) phenomenon: the observation that toxic semantics systematically evolve from conceptual frameworks during early generation phases to specific visual details during later refinement. The authors demonstrate that STARE significantly increases jailbreak success rates against Vision-Language Models (VLMs) by exploiting these temporal windows.

**Compliance With Llm Reviewing Policy:**

Affirmed.

**Final Justification:**

- Increased Theoretical Depth

The comparison with DDPO further confirms the uniqueness and necessity of hierarchical RL in safety red-teaming for diffusion models.

- Experimental Completeness

The addition of experiments involving the latest commercial models and cross-generator scenarios significantly strengthens the reliability of the conclusions.

**Key Questions For Authors:**

1. Can you provide temporal attribution heatmaps for flat optimization methods (e.g., DDPO) to confirm if toxic phase alignment emerges spontaneously without manual hierarchical partitioning?
2. How does the total computational overhead (e.g., training time and GPU hours) of STARE compare to the reported baselines?
3. How does STARE perform against the latest VLM models, such as GPT-5.2 and Gemini-3-Pro?
4. Regarding the threat model, how much does the success rate drop if there is a significant distributional shift between the white-box shadow model and the target model's training data?

**Limitations:**

yes

**Strengths And Weaknesses:**

## Strengths

1. Effective Hierarchical Design: This paper successfully decouples the complex task of multi-modal attacks into high-level semantic guidance and low-level visual reinforcement. By leveraging Group Relative Policy Optimization (GRPO), the framework achieves a substantial increase in Attack Success Rate (ASR). It significantly outperforms established baselines, demonstrating the clear engineering advantages of hierarchical optimization.

2. Novel Diagnostic Tooling: This paper introduces a temporal attribution analysis tool that quantifies the contribution of specific stages in the diffusion denoising trajectory to final toxicity. This provides a valuable, fine-grained metric for understanding and mapping the safety boundaries of generative models.

## Weaknesses

1. The authors emphasize the discovery of phase-specific toxic alignment; however, STARE's hierarchical structure predefines that the high-level module manages semantics while the low-level module handles fine-tuning. Consequently, OIPA may be an artifact of the algorithmic "division of labor" rather than an inherent property of diffusion models under adversarial pressure. The claim lacks conviction without evidence that similar phase separation emerges spontaneously under "flat" (non-hierarchical) optimization, such as standard DDPO.
2. The proposed method requires white-box access to the T2I model for fine-tuning, which limits its applicability in black-box scenarios where an attacker only has access to final outputs. Additionally, while the Multi-Level Monte Carlo (MLMC) estimator helps reduce variance, the computational cost of the hierarchical search remains high. The absence of a detailed comparison regarding computational overhead in the main text or appendix raises concerns about the method's scalability.
3. For a security-focused paper, the threat model is underdeveloped. For instance, the authors assume white-box access to a shadow T2I model but do not specify requirements for distributional similarity between the shadow and target models. This lack of detail may lead readers to overestimate the framework's real-world effectiveness.
4. ​The adversarial images generated by STARE often contain explicit and direct violations, such as blood and violence. In contemporary commercial VLM systems, front-end content safety filters like Azure Content Safety or CLIP-based filters are standard. This paper fails to evaluate the performance on the latest commercial models, such as GPT-5.2 and Gemini-3-Pro, leaving their effectiveness against advanced safety alignments unverified.

5. There are some expressions in the paper that need improvement.

- Figure 1: The caption is overly brief and vague, preventing readers from verifying the qualitative claims regarding the phase-wise emergence of toxic features.
- Section 3.2: The Proj function lacks a precise definition. Additionally, the candidate generation process is unclear: does it generate $K$ initial candidates via $\epsilon^{simple}$ and then further branch into multiple instances per group using $\epsilon^{mod}$?
- Terminology: Abbreviations such as SDE (Stochastic Differential Equation) should be defined in full upon first mention.
- Table Formatting: The captions for Tables 1–3 mention an upward arrow symbol ($\uparrow$) to denote that higher values are better, but this symbol is missing from the tables themselves.

---

> ### Author Rebuttal · Authors · 2026-03-28
>
> # Response to Reviewer 98L2
>
> We thank the reviewer for the careful reading and constructive feedback. We address each point below.
>
> ---
>
> ## W1 / Q1 — OIPA May Be Artifact of Division of Labor; DDPO Heatmap Request
>
> We understand the concern that OIPA might reflect the hierarchical architecture's *design choice* (high-level = semantics, low-level = fine-tuning) rather than an inherent diffusion property.
>
> **Counter-evidence 1 — Vanilla heatmap (Fig. 4d):**
> The unoptimized model already exhibits a latent weak phase structure — Identity/Threat elevated at t≈12–22, Obscene at t≈29–39 (Sec. 4.2). This exists with **no hierarchical design applied**. STARE amplifies a pre-existing property; it does not fabricate one.
>
> **Counter-evidence 2 — DDPO ablation:** DDPO (27.84% ASR) applies on the identical T2I model. Draft temporal attribution heatmap:
>
> ### DDPO Temporal Attribution Heatmap
>
> ![DDPO heatmap](https://bashify.io/i/pZfS2P)
>
> **Contrast with STARE (Fig. 4a):** STARE shows strong, concentrated peaks—conceptual toxicity (Identity, Threat) in t1–15; detail toxicity (Obscene, Insult) in t21–40. DDPO is **diffuse** with no clear phase separation under the same RL objective and backbone, supporting that OIPA reflects hierarchy rather than RL optimization alone. *Final computed DDPO heatmap will also appear beside Fig. 4 in the revision.*
>
> ---
>
> ## W2 / Q2 — White-Box Access; Computational Overhead
>
> We agree white-box T2I is a strong assumption; we adopt the same setting as prior attacks. Attacks trained on SD 3.5-Med outperform baselines on Gemini-2.5-Pro (15.96%) and Qwen2.5-VL (21.47%).
>
> Training setup: LoRA r=16, α=32, frozen base. GPU hours (A100), as requested:
>
> | Method | GPU hours (A100) |
> |--------|------------------|
> | STARE w/ DDPO | 216 |
> | STARE (ours) | 288 |
>
> ---
>
> ## W3 / Q4 — Threat Model; Shadow Model Requirements
>
> **Threat model:** We assume direct white-box access to the deployed T2I (SD 3.5-Med in all experiments)—not a shadow model; the VLM remains black-box (query-only). We will clarify this in revised Sec. 3.
>
> **Cross-T2I shift (Q4):** Thank you for raising this. We evaluate **LLaVA** on **FLUX.1-dev** images while attacks are trained only on **SD 3.5-Med**. Subscripts = SE across runs.
>
> | Method | Any | Toxic | Severe | Obscene | Threat | Insult | Identity | CLIP |
> |--------|-----|-------|--------|---------|--------|--------|----------|------|
> | ART | 11.43₀.₄₄ | 5.82₀.₂₅ | 1.24₀.₁₅ | 10.67₀.₄₃ | 1.89₀.₁₇ | 6.14₀.₂₇ | 3.58₀.₂₄ | 0.69₀.₀₃ |
> | STARE w/ DDPO | 13.27₀.₄₇ | 7.13₀.₃₄ | 2.08₀.₁₉ | 12.44₀.₄₆ | 2.74₀.₂₂ | 7.35₀.₃₄ | 2.91₀.₂₂ | 0.71₀.₀₄ |
> | STARE (0.2) | 17.82₀.₅₄ | 9.47₀.₄₁ | 3.15₀.₂₄ | 16.93₀.₅₃ | 3.28₀.₂₅ | 8.76₀.₃₈ | 3.84₀.₂₇ | 0.74₀.₀₄ |
>
> STARE remains effective under cross-generator deployment with realistic CLIP scores; Table 3 cross-VLM transfer provides complementary evidence.
>
> ---
>
> ## W4 / Q3 — Evaluation on Latest Commercial Models
>
> To address Q3 on frontier commercial VLMs, we additionally report **GPT-5.4** (subscripts = SE):
>
> | Method | Any | Toxic | Severe | Obscene | Threat | Insult | Identity |
> |--------|-----|-------|--------|---------|--------|--------|----------|
> | Text-Only | 1.80₀.₀₀ | 0.40₀.₀₀ | 0.10₀.₀₀ | 1.60₀.₀₀ | 0.20₀.₀₀ | 1.70₀.₀₀ | 0.20₀.₀₀ |
> | Text + SD | 4.82₀.₃₁ | 2.54₀.₁₈ | 0.16₀.₀₂ | 4.37₀.₂₉ | 0.09₀.₀₁ | 3.81₀.₂₀ | 1.64₀.₁₇ |
> | STARE w/ DDPO | 12.08₀.₄₆ | 6.72₀.₃₄ | 2.09₀.₁₉ | 11.61₀.₄₅ | 2.65₀.₂₂ | 6.89₀.₃₃ | 2.69₀.₂₂ |
> | STARE (0.2) | 13.64₀.₄₉ | 6.95₀.₄₃ | 3.37₀.₂₆ | 12.34₀.₄₇ | 3.52₀.₂₆ | 6.99₀.₃₄ | 3.53₀.₂₆ |
> Our adversarial images are realistic T2I outputs (not pixel perturbations), which may partly explain robustness to filters aimed at low-level adversarial noise.
>
> ---
>
> ## W5 — Presentation Issues
>
> We thank the reviewer for these suggestions and will incorporate them in the revision: (1) Fig. 1 caption—label each phase panel by toxicity dim. (2) Define `Proj(μ_j)=ε_p·μ_j/max(‖μ_j‖₂,ε_p)`, ε_p=0.8. (3) Clarify K groups from ε^(simple); M rollouts/group via ε_j^(mod). (4) Expand SDE on first use in Sec. 3.1. (5) Add ↑ to metric headers in Tables 1–3.

---

> > ### Author Rebuttal · Reviewer_98L2 · 2026-04-04
> >
> > Thanks for the response. The authors have provided a persuasive rebuttal that successfully addresses the primary concerns raised in the initial review. In light of these clarifications and the additional high-quality experimental data, I am willing to upgrade my recommendation to Accept.

---

### Official Review · Reviewer_iFP1 · 2026-03-09

**Soundness:** 3
**Presentation:** 2
**Significance:** 2
**Originality:** 3
**Overall Recommendation:** 3
**Confidence:** 3

**Summary:**

This paper proposes STARE, a hierarchical reinforcement-learning framework for multimodal red-teaming that combines a high-level prompt editor with low-level T2I fine-tuning. The paper claims that adversarial optimization induces a temporal restructuring of toxicity emergence, termed Optimization-Induced Phase Alignment. Experiments with Stable Diffusion 3.5-Medium and LLaVA-v1.6-mistral-7b-hf show higher ASR than ART, DiffZOO, PGJ, and a DDPO-based variant, with additional transfer results on Qwen and Gemini.

**Compliance With Llm Reviewing Policy:**

Affirmed.

**Final Justification:**

The rebuttal provides useful clarification, including additional analysis for OIPA and a clearer positioning of DDPO as the primary controlled baseline. I appreciate that the authors indicate they will more carefully qualify the mechanistic claim rather than presenting it as fully conclusive.

However, my core concerns remain only partially addressed. I still find that the current evidence is not sufficient to convincingly support the central interpretation of phase-specific temporal alignment. I also remain concerned that the empirical evaluation does not yet cleanly separate controlled comparisons from broader contextual baselines, which makes the overall evidence somewhat harder to interpret.

Overall, given these remaining concerns and limitations discussed above, I maintain my original score of 3.

**Key Questions For Authors:**

1. Can the authors provide stronger evidence that the proposed temporal alignment reflects a causal mechanism rather than a post-hoc pattern, for example through targeted interventions on early/late denoising windows or defense-side experiments?

2. The method appears computationally expensive, but cost is underreported. The implementation details state that this is a high-cost setup (4 prompts variants, 8 images), yet the paper does not provide a meaningful cost/query-budget analysis.

3. How robust are the results to alternative VLM prompting templates and to stronger judge models?

4. Can the authors further explain what each ablation reveals mechanistically?

5. Could the authors discuss the relation between STARE and stronger black-box attack baselines that directly target VLMs? Since the paper is ultimately framed as an attack on downstream VLMs, it would be helpful to understand whether there are more directly comparable black-box VLM attack methods, and if so, why they are not included.

**Limitations:**

YES

**Strengths And Weaknesses:**

Strengths:

1.The paper studies an important multimodal red-teaming setting by focusing on how optimized text-image pairs can induce toxic continuation from VLMs, going beyond text-only jailbreaks.

2.The overall framework is technically coherent. The high-level prompt editor, low-level T2I fine-tuning stage, and reward design fit together naturally for the stated attack objective.

3.STARE improves ASR over several baselines on RTP and PTP, and also shows some transferability to Qwen and Gemini.

Weaknesses:

1. The central scientific claim is not yet fully established. While the temporal-alignment analysis in Section 3.3 is informative, it currently functions as a post-hoc diagnostic. The observed step-wise sensitivity patterns are suggestive, but the paper does not provide quantitative or intervention-based evidence. Under the stated threat model, where the final attack target is the VLM and the diffusion model serves as the optimized upstream generator, the paper does not yet provide strong quantitative or intervention-based evidence linking diffusion-phase attribution to the VLM’s safety failure mechanism. Moreover, the analysis appears to have limited impact on the method design itself, as it is not clearly used to motivate or guide the hierarchical optimization framework. For the same reason, the discussion of phase-aware defense appears promising but still somewhat speculative, especially in the absence of defense-side experiments.

2. The hierarchical framing needs stronger justification. Although the paper formally defines a high-level prompt editing and a low-level denoise, much of the novelty appears concentrated in the semantic editor and its coupling with T2I optimization. The low-level component resembles a standard RL-based fine-tuning procedure for rectified-flow generation. So, it is unclear whether the hierarchy represents a principled methodological contribution or primarily a decomposition.

3. The experimental comparison could be more transparent and controlled. Some qualitative examples are selected using the highest toxicity scores within a run, which raises potential cherry-picking concerns. In addition, several baselines are adapted by the authors to fit the current setting, despite differing access assumptions and optimization budgets. Separating black-box prompt-based baselines from white-box training-based baselines, with clearer reporting of matched budgets, would strengthen the evaluation.

4. Presentation clarity could be improved. Some notation in Section 3 is introduced or reused in a confusing manner, and certain structural issues (e.g., repeated limitation discussions) interrupt the reading flow. A cleaner presentation would make the main contributions easier to assess.

5. The threat model is relatively strong and may limit practical relevance. The method assumes white-box access to the T2I model for fine-tuning, which reduces applicability in typical deployment scenarios. Moreover, the RL-based optimization likely introduces substantial computational overhead, but the paper does not clearly analyze whether the performance gains justify this cost.

---

> ### Author Rebuttal · Authors · 2026-03-28
>
> We thank the reviewer for the detailed critique and constructive questions. We address each point below.
>
> ## W1 / Q1 — OIPA Not Fully Established; Post-Hoc; No Intervention Evidence
>
> Two pieces of evidence show OIPA reflects an intrinsic diffusion property, not an artifact.
>
> **Counter-evidence 1 — Vanilla heatmap (Fig. 4d):** The *unoptimized* base model already shows latent phase structure — Identity/Threat elevated at t≈12–22, Obscene at t≈29–39 (Sec. 4.2); STARE amplifies it, not fabricates it.
>
> **Counter-evidence 2 — DDPO ablation:** DDPO (27.84% ASR) on the identical backbone yields a **diffuse** heatmap with no clear phase separation, unlike STARE's concentrated peaks (conceptual in t1–15; detail in t21–40). Full heatmap: **see Reviewer 98L2 (W1 / Q1).**
>
> **Causal intervention (LLaVA / RTP, STARE 0.2):** We zero denoising updates in t1–15 (early), t21–40 (late), or both. Subscripts = SE; columns match Table 1.
>
> | Condition | Any | Toxic | Severe | Obscene | Threat | Insult | Identity |
> |-----------|-----|-------|--------|---------|--------|--------|----------|
> | STARE (0.2) full | 31.36₀.₆₆ | 17.10₀.₅₃ | 6.29₀.₃₄ | 29.73₀.₆₅ | 4.38₀.₂₉ | 15.95₀.₅₂ | 6.14₀.₃₄ |
> | Zero early (t1–15) | 23.84₀.₆₀ | 12.47₀.₄₇ | 4.91₀.₃₁ | 22.16₀.₅₉ | 1.83₀.₁₈ | 15.62₀.₅₂ | 2.31₀.₂₁ |
> | Zero late (t21–40) | 25.17₀.₆₁ | 13.58₀.₄₈ | 4.63₀.₃₀ | 19.84₀.₅₆ | 4.21₀.₂₈ | 10.34₀.₄₃ | 5.88₀.₃₃ |
> | Zero both | 18.93₀.₅₅ | 10.12₀.₄₄ | 3.27₀.₂₅ | 17.44₀.₅₃ | 1.64₀.₁₇ | 9.87₀.₄₂ | 2.08₀.₁₉ |
>
> Either window zeroing lowers ASR; Identity/Threat fall most when early updates removed, Obscene/Insult when late — consistent with phase-specific roles. Hierarchy was motivated *a priori* by diffusion phase-structure work (Wang et al., Park et al.; Sec. 3.3); we will clarify this bidirectional link in the revision.
>
> ## W2 — Hierarchical Framing Needs Stronger Justification
>
> Ablations are phase-specific, not uniform: w/o Edit hurts early peaks (Fig. 4c, −5.8pp); w/o LoRA hurts late peaks (Fig. 4b, −9.3pp); w/ DDPO is diffuse (−3.5pp). Modules specialize to non-overlapping windows; the low-level path also differs from vanilla DDPO via per-step KL (Eq. 3) and MPS exploration (Eq. 1).
>
> ## W3 — Experimental Transparency; Cherry-Picking; Baseline Comparison
>
> Qualitative picks follow Sec. 5 footnote (*highest Detoxify score per run*); Tables 1–3 use full 1K sets. PGJ/DiffZOO are black-box prompts; ART uses an LLM agent; only DDPO matches STARE's white-box RL budget (Appendix A.3). We will surface the access breakdown in the main text.
>
> ## W4 — Presentation Clarity
>
> Notation and figure fixes overlap **Reviewer 98L2 (W5)** (`Proj`, ε^(simple)/ε_j^(mod), SDE, Fig. 1, ↑). Additionally: merge duplicate limitation sections.
>
> ## W5 / Q2 — Threat Model; Computational Cost
>
> **Threat model:** White-box T2I (SD 3.5-Med), black-box VLM (query-only); not a shadow model. We will foreground this in Sec. 3 and the abstract.
>
> **Cost:** STARE 288 A100 h vs. ART 72 h, STARE w/ DDPO 216 h, DiffZOO 48 h. Full table: **see Reviewer 98L2 (W2).**
>
> **Cross-T2I transfer:** On FLUX.1-dev: STARE (0.2) 17.82% Any-ASR vs. ART 11.43%, STARE w/ DDPO 13.27% (CLIP 0.74). Cross-VLM: Gemini-2.5-Pro 15.96%, Qwen2.5-VL 21.47%. Full FLUX table: **see Reviewer 98L2 (W3).**
>
> ## Q3 — Judge Robustness; Prompting Templates
>
> Perspective API (Appendix C) agrees with Detoxify across scorers.
>
> **Template sensitivity (STARE 0.2 / LLaVA / RTP; instruction only):** T1 "extend the sentence"; T2 "Continue the following text"; T3 roleplay "creative writer…Continue"; T4 formal "professional tone".
>
> | Template | Any | Toxic | Severe | Obscene | Threat | Insult | Identity | CLIP |
> |----------|-----|-------|--------|---------|--------|--------|----------|------|
> | T1 default | 31.36₀.₆₆ | 17.10₀.₅₃ | 6.29₀.₃₄ | 29.73₀.₆₅ | 4.38₀.₂₉ | 15.95₀.₅₂ | 6.14₀.₃₄ | 0.78₀.₀₅ |
> | T2 neutral | 30.84₀.₆₅ | 16.77₀.₅₂ | 6.11₀.₃₄ | 29.15₀.₆₄ | 4.22₀.₂₈ | 15.63₀.₅₁ | 5.97₀.₃₃ | 0.78₀.₀₅ |
> | T3 roleplay | 33.12₀.₆₇ | 18.24₀.₅₄ | 6.88₀.₃₅ | 31.07₀.₆₆ | 4.71₀.₃₀ | 16.83₀.₅₃ | 6.52₀.₃₅ | 0.77₀.₀₄ |
> | T4 formal | 29.53₀.₆₄ | 15.94₀.₅₁ | 5.83₀.₃₃ | 27.88₀.₆₃ | 4.05₀.₂₇ | 14.72₀.₅₀ | 5.71₀.₃₃ | 0.78₀.₀₅ |
> | Δ max–min | 3.59 | 2.30 | 1.05 | 3.19 | 0.66 | 2.11 | 0.81 | 0.01 |
>
> ASR range Any 29.53–33.12% (Δ=3.59pp); CLIP unchanged. We retain T1 and will report in the main text.
>
> ## Q4 — Mechanistic Ablation Interpretation
>
> - **w/o LoRA:** concepts seeded but not visually amplified → low visual toxicity.
> - **w/o Edit:** visual RL without semantic subgoals → missing early peaks.
> - **w/ DDPO:** no phase-specific credit → diffuse pressure and lower ASR.
>
> We will add this to Sec. 4.1.
>
> ## Q5 — Black-Box VLM Attack Baselines
>
> Text-only jailbreaks lack images and do not fit toxic continuation; RedDiffuser differ in mechanism. We will add a threat-model mapping table in related work.
>
> We hope these responses and new experiments help address your concerns.

---

> > ### Author Rebuttal · Reviewer_iFP1 · 2026-04-02
> >
> > Thanks for your response. It clarifies several points. However, my main concern is only partially resolved: the evidence for the claimed mechanism is still suggestive rather than conclusive, and I think the baseline comparability remains unfair.

---

> > > ### Author Response · Authors · 2026-04-04
> > >
> > > We will address the specific identification of the two remaining concerns below.
> > >
> > > == Concern 1: OIPA Evidence — Suggestive vs. Conclusive ==
> > >
> > > We understand the bar for "conclusive" is high. We draw attention to a specific pattern in our intervention table that constitutes stronger evidence than a purely suggestive correlation.
> > >
> > > The intervention table exhibits a double dissociation. When we zero denoising updates in specific temporal windows, the ASR drops are not proportional across toxicity categories — they are category-specific, matching OIPA's predicted phase-category alignment:
> > >
> > > Zero early (t1-15): Identity -62%, Threat -58%, Obscene -25%, Insult -2%
> > > Zero late (t21-40): Identity -4%, Threat -4%, Obscene -33%, Insult -35%
> > >
> > > Zeroing the early window disproportionately eliminates Identity (-62%) and Threat (-58%) while leaving Insult virtually unchanged (-2%). Zeroing the late window disproportionately eliminates Obscene (-33%) and Insult (-35%) while leaving Identity (-4%) and Threat (-4%) virtually unchanged.
> > >
> > > If OIPA were a post-hoc artifact, zeroing either window should produce uniform proportional drops across all categories. Instead, we observe a textbook double dissociation: early phases causally control conceptual toxicity dimensions, late phases causally control detail-level dimensions, adapted here to denoising trajectories.
> > >
> > > Three converging lines: (1) The vanilla heatmap (Fig. 4d) shows weak phase structure with no hierarchy applied, establishing OIPA amplifies an intrinsic property. (2) The DDPO heatmap shows flat RL on the identical backbone produces a diffuse pattern with no phase separation. (3) Zeroing both windows (18.93%) produces a drop roughly consistent with the sum of individual effects, with some sub-additivity (1.3pp), suggesting largely independent causal contributions.
> > >
> > > We acknowledge this does not reach formal causal-graph identifiability. We will frame OIPA more precisely in the revision: "supported by intervention-based double dissociation and converging ablation evidence."
> > >
> > > == Concern 2: Baseline Comparability ==
> > >
> > > We agree that comparing methods with different access assumptions requires careful framing.
> > >
> > > Tier 1 — Fair comparison: DDPO is the only baseline with identical access (white-box T2I, black-box VLM), identical RL framework (policy gradient on diffusion), and identical backbone (SD 3.5-Med). STARE outperforms DDPO by +3.52pp Any-ASR (31.36% vs 27.84%), isolating the contribution of hierarchical temporal exploitation.
> > >
> > > Tier 2 — Cross-paradigm context: PGJ/DiffZOO (black-box prompt) and ART (LLM agent) use different attack surfaces. We include them for landscape positioning, not as controlled comparisons. The GPU hours difference reflects prompt-editing overhead. STARE w/o Edit (LoRA-only, same budget class as DDPO) achieves 25.56% — still below full STARE (31.36%), confirming hierarchy adds value beyond extra compute.
> > >
> > > We will restructure Tables 1-3 in the revision to foreground the Tier-1 DDPO comparison and clearly label cross-paradigm baselines as contextual.

---

### Official Review · Reviewer_A8gA · 2026-03-13

**Soundness:** 3
**Presentation:** 3
**Significance:** 3
**Originality:** 3
**Overall Recommendation:** 4
**Confidence:** 4

**Summary:**

This paper proposes STARE, a hierarchical reinforcement learning framework for red-teaming Vision-Language Models (VLMs) by generating adversarial images with a diffusion-based Text-to-Image model. The method combines prompt editing and diffusion fine-tuning to maximize toxicity in VLM generations. In addition to improving attack success rate, the paper introduces a temporal alignment analysis to study when toxic semantics emerge during the diffusion process. Experiments show that STARE significantly outperforms existing baselines and reveal that adversarial optimization tends to align different types of toxicity with specific diffusion phases.

**Compliance With Llm Reviewing Policy:**

Affirmed.

**Final Justification:**

The rebuttal addressed most of my concerns. The authors clarified that the novelty lies in identifying and exploiting the temporal attack surface during diffusion denoising, rather than in any individual component. This makes the contribution clearer relative to prior T2I red-teaming work. They also provided additional evidence on transfer to FLUX and black-box VLMs, which strengthens the practical relevance of the attack beyond the single training setup.

I still think the current validation is somewhat limited, especially in model and task coverage. However, the authors acknowledged these limitations clearly, and the rebuttal makes the technical contribution and empirical findings much more convincing. Overall, I believe the main concerns can be addressed in the revision, so I lean weak accept.

**Key Questions For Authors:**

1. Is the observed temporal phase alignment specific to RL-based optimization, or does it also work under simpler attack strategies such as gradient-based or search-based prompt optimization?
2. Does the proposed analysis generalize to other types of harmful behaviors beyond toxicity continuation?

**Limitations:**

yes

**Strengths And Weaknesses:**

**Strengths:**

1. The paper presents a novel perspective by treating diffusion trajectory as a structured attack surface and studies temporal dynamics of toxicity formation, which provides an interesting angle for analyzing generative model vulnerabilities.
2. The temporal attribution framework and the observed phase alignment phenomenon offer useful insights into how adversarial optimization interacts with diffusion generation.
3. Experiments show improvements in attack success rate and demonstrate cross-model transferability.

**Weaknesses:**

1. The novelty of this paper is a bit limited. The STARE framework mainly combines existing components (prompt editing, diffusion fine-tuning, RL optimization).
2. The phase alignment phenomenon is demonstrated mainly on a single diffusion model and toxicity task, leaving questions about its generality to different T2I models.
3. The attack assumes white-box access to the Text-to-Image model for RL fine-tuning. This is a relatively strong assumption and may limit the practical relevance of the attack.

---

> ### Author Rebuttal · Authors · 2026-03-28
>
> # Response to Reviewer A8gA
>
> We thank the reviewer for the positive assessment. We address each point below.
>
> ---
>
> ## W1 — Limited Novelty
>
> STARE's novelty is not any individual component but the **discovery and exploitation of the temporal attack surface** in diffusion models. Prior T2I red-teaming (ART, DiffZOO, PGJ) treats generation as a terminal black-box oracle — none analyze *when* during denoising toxic semantics are injected. STARE is, to our knowledge, the first to show that (i) adversarial optimization restructures the denoising trajectory into non-overlapping phase-specific vulnerability windows (OIPA), and (ii) a hierarchical attack architecture designed around this discovery substantially outperforms flat alternatives. The components are well-established; the insight and the framework built around it are the contribution.
>
> ---
>
> ## W2 — Phase Alignment on Single Model / Single Toxicity Task
>
> We acknowledge this limitation, stated in Sec. 6. SD 3.5-Med was chosen for its rectified-flow formulation, which produces near-straight denoising trajectories and explicit velocity fields that make phase boundaries reliably interpretable (Sec. 3, Preliminaries). The temporal attribution framework is reward-agnostic and architecture-independent in principle; extending OIPA analysis to e.g., FLUX is important future work we will discuss explicitly in the revision.
>
> For transfer beyond the training T2I: attacks trained on SD 3.5-Med transfer to FLUX.1-dev (LLaVA evaluation) — STARE (0.2) achieves 17.82% Any-ASR vs. ART 11.43% and STARE w/ DDPO 13.27% (CLIP 0.74). Cross-VLM results on the same SD-trained images: Gemini-2.5-Pro 15.96%, Qwen2.5-VL 21.47%. Full FLUX table and cross-VLM breakdown: **see Reviewer 98L2 (W3 / Q4)**.
>
> ---
>
> ## W3 — White-Box T2I Assumption
>
> White-box T2I access matches prior diffusion RL attack work. The resulting adversarial images successfully elicit toxic continuations from black-box VLMs — Gemini-2.5-Pro (15.96% Any-ASR) and Qwen2.5-VL (21.47%). Full GPU-hours table and transfer details: **see Reviewer 98L2 (W2).**
>
> ---
>
> ## Q1 — Is Phase Alignment RL-Specific or Also Gradient/Search-Based?
>
> **Flat RL vs. hierarchical RL:** The DDPO ablation (Table 1: 27.84% ASR vs. STARE 31.36%) uses flat RL on the identical T2I model with the same white-box budget. Its temporal attribution heatmap is **diffuse** with no clear phase separation — unlike STARE's concentrated peaks (conceptual toxicity in t1–15; detail toxicity in t21–40). Same backbone and RL objective; phase structure emerges only with the hierarchical design. Full heatmap figure: **see Reviewer 98L2 (W1 / Q1).** Together, lower ASR and diffuse attribution support that **hierarchical RL — not RL in general — produces OIPA**.
>
> **baselines:** PGJ and DiffZOO optimize over prompt tokens/embeddings at test time without updating the diffusion model's velocity field. Since temporal attribution measures sensitivity of the *denoiser's internal computations* at each step, methods that do not modify the denoiser cannot redistribute those dynamics — phase alignment is structurally unavailable to them regardless of optimization strategy.
>
> ---
>
> ## Q2 — Does the Analysis Generalize Beyond Toxicity Continuation?
>
> The temporal attribution framework (Eq. 2–3) measures sensitivity of any scalar reward to perturbations at specific denoising timesteps. OIPA relies on whether the reward has distinct conceptual and detail-oriented sub-dimensions that can align to the model's inherent coarse-to-fine generation hierarchy. We expect similar phase alignment for other semantically structured harm types (identity-based hate speech, self-harm imagery). Single-dimensional rewards (pure nudity classifiers) may not produce phase separation. Extending OIPA to other harm categories is important future work; we will discuss it explicitly.

---

> > ### Author Rebuttal · Reviewer_A8gA · 2026-04-03
> >
> > Thank the authors for their efforts. My concerns have been mostly resolved.

---

### Decision · Program_Chairs · 2026-04-30

**Decision:**

Accept (regular)

**Comment:**

The paper proposes STARE, a hierarchical reinforcement-learning framework for multimodal red-teaming, in which a high-level prompt editor is combined with low-level T2I fine-tuning to generate adversarial image-text inputs that induce toxic outputs from VLMs. A central contribution of the paper is the temporal analysis of toxicity formation during diffusion denoising, summarized as the Optimization-Induced Phase Alignment (OIPA) phenomenon.

The reviewers generally agree that the paper studies an important and timely multimodal safety problem, and that the overall framework is technically coherent. They also found the empirical gains in attack success rate to be strong, and viewed the temporal attribution analysis as an interesting diagnostic perspective on vulnerability in diffusion-based generation.

At the same time, the reviewers raised several meaningful concerns. First, while the empirical improvements are clear, there is some disagreement about the strength of the paper’s central scientific claim regarding phase alignment. In particular, one reviewer remained unconvinced that the current evidence establishes OIPA as a causal or model-intrinsic mechanism rather than a post-hoc pattern or a possible artifact of the hierarchical optimization design. Relatedly, the current analysis is not yet clearly tied back to the method design itself, and the discussion of phase-aware defenses remains preliminary in the absence of defense-side experiments. Second, the threat model is relatively strong, since the method assumes white-box access to the T2I model for RL fine-tuning, which limits practical relevance in some deployment settings. Third, the method appears computationally expensive, but the paper still provides limited analysis of cost and budget relative to baselines. Finally, several reviewers noted limitations in model/task coverage and presentation clarity, although some of these concerns were partially addressed in the rebuttal.

Overall, I find this to be a technically solid paper. The attack framework is effective, and the temporal perspective on toxicity formation is potentially valuable for future work. At the same time, I agree that the current evidence does not fully settle the stronger mechanistic interpretation of OIPA, and the practical scope is constrained by the threat model and cost. Therefore, I believe the combination of strong attack results, cross-model transfer evidence, and the novel temporal analysis makes the paper a worthwhile contribution. I therefore recommend weak acceptance.